# Disrupted Endoplasmic Reticulum Ca^2+^ Handling: A Harβinger of β-Cell Failure

**DOI:** 10.3390/biology13060379

**Published:** 2024-05-25

**Authors:** Jordyn R. Dobson, David A. Jacobson

**Affiliations:** Department of Molecular Physiology and Biophysics, Vanderbilt University, Nashville, TN 37232, USA; jordyn.dobson@vanderbilt.edu

**Keywords:** β-cell, islet, diabetes, ER stress, Ca^2+^ handling, unfolded protein response, insulin, chaperone, ryanodine receptor, inositol trisphosphate receptor

## Abstract

**Simple Summary:**

Diabetes results from insufficient insulin production and secretion. Insulin accounts for almost half of the protein produced in the pancreatic β-cell, which occurs in the endoplasmic reticulum (ER). Ca^2+^ is a key signal within the ER that controls insulin production and processing. β-cell ER Ca^2+^ (Ca^2+^_ER_) depletion during the pathogenesis of type 1 and type 2 diabetes leads to reduced insulin production and secretion. Mutations in insulin that cause monogenic diabetes also reduce Ca^2+^_ER_ and result in β-cell dysfunction. Thus, the mechanisms that tune β-cell Ca^2+^_ER_ and become disrupted in diabetes are potential targets for enhancing insulin production and reducing hyperglycemia. This review focuses on β-cell Ca^2+^_ER_ and how it impacts β-cell function and dysfunction.

**Abstract:**

The β-cell workload increases in the setting of insulin resistance and reduced β-cell mass, which occurs in type 2 and type 1 diabetes, respectively. The prolonged elevation of insulin production and secretion during the pathogenesis of diabetes results in β-cell ER stress. The depletion of β-cell Ca^2+^_ER_ during ER stress activates the unfolded protein response, leading to β-cell dysfunction. Ca^2+^_ER_ is involved in many pathways that are critical to β-cell function, such as protein processing, tuning organelle and cytosolic Ca^2+^ handling, and modulating lipid homeostasis. Mutations that promote β-cell ER stress and deplete Ca^2+^_ER_ stores are associated with or cause diabetes (e.g., mutations in ryanodine receptors and insulin). Thus, improving β-cell Ca^2+^_ER_ handling and reducing ER stress under diabetogenic conditions could preserve β-cell function and delay or prevent the onset of diabetes. This review focuses on how mechanisms that control β-cell Ca^2+^_ER_ are perturbed during the pathogenesis of diabetes and contribute to β-cell failure.

## 1. Introduction

Disruption in the production and/or secretion of insulin contribute(s) to hyperglycemia during the pathogenesis of diabetes. The pancreatic β-cell Ca^2+^ serves critical roles not only in stimulating insulin secretion but also in endoplasmic reticulum (ER) production and the processing of insulin. Due to the constant production and high daily requirement of insulin, the ER of the β-cell has a finely tuned quality control system that is modulated by Ca^2+^. The total concentration of Ca^2+^_ER_ is ~2 mM, with the majority buffered by Ca^2+^_ER_ binding proteins such as chaperones. The free Ca^2+^_ER_ ranges between 50 and 500 µM (compared to 0.1–2 µM in the cytosol), which sets up a large driving force for Ca^2+^_ER_ release [1,2,3]. Stressful conditions associated with diabetes reduce Ca^2+^_ER_ stores; this elevates ER stress and initiates the unfolded protein response (UPR), which initially protects β-cells from diabetogenic stress but also limits insulin production and secretion [4,5]. The prolonged disruption of Ca^2+^_ER_ can eventually lead to β-cell failure and/or senescence [6]. Importantly, alterations in the levels and/or function of the proteins involved in controlling Ca^2+^_ER_ homeostasis can lead to diabetic phenotypes due to β-cell dysfunction. Moreover, mutations that disrupt insulin folding and promote accumulation in the ER lead to reduced Ca^2+^_ER_ storage, UPR, and β-cell failure [2,7,8]. Therefore, elevating Ca^2+^_ER_ under diabetic conditions helps to restore β-cell function and glucose homeostasis [9,10]. This review focuses on how β-cell Ca^2+^_ER_ becomes disrupted and contributes to diabetes pathogenesis by impairing insulin production, secretion, and quality control (Figure 1).

### 1.1. Ca^2+^_ER_ in β-Cell Survival and Health

The ER serves a critical role in β-cell pro-survival mechanisms for recovering from the stress associated with the high production of insulin. The necessity of Ca^2+^_ER_ for β-cell survival is demonstrated by increased apoptosis following the prolonged inhibition of Ca^2+^_ER_ uptake [11]. However, the activation of the adaptive UPR following the acute depletion of β-cell Ca^2+^_ER_ stores enables protection from stress, which is critical to β-cell survival and longevity. This adaptive response includes the upregulation of IRE1, PERK, and ATF6 signaling, which limit translation, increase chaperone expression, and eliminate misfolded proteins [12]. The prolonged depletion of Ca^2+^_ER_ stores leads to terminal UPR activation and subsequent pro-apoptotic signaling. This includes the degradation of ER-localized mRNAs by IRE1α, the increased expression of C/EBP homologous protein (CHOP), and the activation of caspase 2-mediated apoptosis [13]. Thus, Ca^2+^_ER_ homeostasis serves critical roles in β-cell survival and adaptation to stress. Interestingly, β-cells from nondiabetic donors transition between different states of UPR activation, which are associated with alterations in insulin biosynthesis. Transient elevations in UPR likely allow for β-cell rest and recovery from stress associated with insulin production and secretion [14]. Therefore, the Ca^2+^_ER_ control of UPR provides a pro-survival mechanism that allows for the β-cell to maintain health under stressful conditions, which can be recurrent due to high levels of insulin production and secretion.

### 1.2. Ca^2+^_ER_ Control of Insulin Secretion

Insulin secretion is finely tuned by Ca^2+^_ER_ uptake and release. Ca^2+^_ER_ oscillates in phase with glucose-induced cytosolic Ca^2+^ (Ca^2+^_C_) oscillations, contributing to the regulation of Ca^2+^_C_ and pulsatile insulin secretion [15]. Ca^2+^-induced Ca^2+^ release (CICR) from the ER amplifies insulin secretion. This occurs via the Ca^2+^ activation of ryanodine receptors (RyRs) as well as phospholipase C-mediated IP_3_ production and IP_3_R activation [16]. Ca^2+^_ER_ handling not only tunes glucose-stimulated insulin secretion (GSIS) but also stimulates insulin secretion in response to Gq-G protein coupled receptor (Gq-GPCR) signaling via Ca^2+^_ER_ release through IP_3_Rs [17]. Thus, alterations in Ca^2+^_ER_ perturb insulin secretion. For example, the acute inhibition of Ca^2+^_ER_ uptake increases glucose-stimulated Ca^2+^_C_ and insulin secretion, whereas the sustained depletion of Ca^2+^_ER_ stores upregulates the UPR, limiting insulin production and secretion. The machinery that controls Ca^2+^_ER_ and how it modulates insulin secretion and contributes to β-cell dysfunction are discussed in detail below.

## 2. Ca^2+^_ER_ Handling Proteins and Their β-Cell Function(s)

The ER contains specialized Ca^2+^ pumps, ion channels, and Ca^2+^ sensing/regulated proteins that control Ca^2+^_ER_ sequestration, uptake, and release. Ca^2+^_ER_ handling helps shape Ca^2+^_C_ oscillations and pulsatile insulin secretion [15]. In addition, the release of Ca^2+^_ER_ generates a negative charge buildup on the ER membrane. Therefore, ion channels and pumps serve to limit the generation of ER membrane potential, promote the driving force of Ca^2+^_ER_ release, and provide a setpoint for Ca^2+^_ER_ store homeostasis through balanced Ca^2+^ release and refilling. Dysfunction and/or changes in the expression of proteins that modulate Ca^2+^_ER_ impair proper insulin production and secretion, which, when prolonged, induces β-cell failure. The primary mediators of β-cell Ca^2+^_ER_ handling are detailed below.

### 2.1. Sarco/Endoplasmic Reticulum Calcium ATPase (SERCA)

Function: The uptake of Ca^2+^ into the ER occurs via SERCA, a P-type ATPase, which pumps 2 Ca^2+^ ions into the ER lumen and extrudes two to three protons per ATP hydrolyzed. While SERCA is critical for keeping Ca^2+^_ER_ stores elevated, changes in its activity also help fine-tune Ca^2+^_C_ handling (Figure 2). β-cell glucose metabolism leads to the ATP energization of SERCA and Ca^2+^_ER_ uptake, which results in a drop in Ca^2+^_C_ [18,19]. Interestingly, Ca^2+^_ER_ oscillates in phase with glucose-induced Ca^2+^_C_ oscillations [3]. Therefore, Ca^2+^_ER_ uptake occurs during the upstroke of a Ca^2+^ oscillation and Ca^2+^_ER_ release occurs during the termination [3,15,18]. This could be due, in part, to ATP oscillating out of phase with Ca^2+^ oscillations, thus energizing SERCA during each Ca^2+^ oscillation. Furthermore, elevated intraluminal Ca^2+^_ER_ inhibits SERCA and reduces its activity, as Ca^2+^_ER_ stores increase during the Ca^2+^ oscillation, which increases the relative contribution of Ca^2+^_ER_ leak. This is supported by the pharmacological inhibition of SERCA, which causes each Ca^2+^ oscillation to have an accelerated rising phase, increased amplitude, and more rapid termination phase [20]. Ca^2+^ handling is also influenced by the different properties of SERCA2b (*ATP2A2*) and SERCA3a-c (*ATP2A3*) isoforms expressed in β-cells [10,21]. SERCA2b is the most abundant β-cell isoform and has a low affinity for Ca^2+^ (K Ca^2+^ = 0.2 µM), whereas SERCA3 has a lower expression (~50% compared to SERCA2b) and a higher affinity for Ca^2+^ (K Ca^2+^ = 1.2 µM) [22,23]. SERCA2b actively pumps Ca^2+^ into the ER lumen at basal and stimulatory glucose concentrations [18]. On the other hand, SERCA3 does not significantly alter basal Ca^2+^_ER_ uptake due to its higher affinity for Ca^2+^ but contributes to Ca^2+^_ER_ uptake during glucose-induced Ca^2+^ influx [18,24,25]. Taken together, small changes in SERCA activity play an important role in tuning β-cell Ca^2+^ handling. The maintenance of Ca^2+^_ER_ storage by SERCA is also critical for protecting β-cells from stress; thus, prolonged SERCA inhibition leads to β-cell death.

SERCA expression and activity are also regulated by insulin signaling through insulin receptor substrate-1 and -2 proteins (IRS-1 and IRS-2). IRS-1 forms a complex with SERCA3 that can be enhanced with insulin [26]. The overexpression of IRS-1 in β-cells inhibits SERCA3, leading to reduced Ca^2+^_ER_ storage and elevated Ca^2+^_C_ [27,28]. In addition, IRS-1 and IRS-2 signaling enhances the expression of the genes encoding SERCA2 and SERCA3 (*Atp2a2* and *Atp2a3*), which are reduced in IRS-1 or IRS-2 knockout mice [29,30]. IRS-1 knockout mice also show reduced glucose-induced Ca^2+^ influx and elevated Ca^2+^_ER_ stores [29,30]. However, as SERCA inhibitors cause elevated glucose-induced Ca^2+^ influx and reduced Ca^2+^_ER_ storage, the loss of SERCA abundance in IRS knockout models would not be expected to elevate Ca^2+^_ER_ stores. Thus, the loss of IRS interaction and the inhibition of SERCA could potentially cause the observed elevation in Ca^2+^_ER_ stores in islets from either IRS-1 or IRS-2 knockouts [29,30]. The insulin control of Ca^2+^_ER_ through the IRS-mediated modulation of SERCA expression and activity could serve as a positive feedback circuit to enhance Ca^2+^_C_ and insulin secretion [31,32,33].

SERCA activity is further regulated by post-translational modification such as O-GlcNAcylation. The O-GlcNAcylation of islet proteins including SERCA2 is increased under diet-induced hyperlipidemia [34]; this results in the potentiation of insulin secretion. Furthermore, the O-GlcNAcylation of SERCA2 increases its abundance and likely increases its function [34]. Importantly, defects in insulin secretion from O-linked N-acetylglycosamine transferase (OGT) ablation can be rescued by SERCA activation, which suggests that impaired SERCA2 activity may contribute to blunted Ca^2+^ influx in islets with OGT ablation [34,35]. The role of SERCA2 O-GlcNAcylation in Ca^2+^_ER_ storage under physiological and diabetic conditions remains to be determined.

SERCA can also be activated post-translationally in response to glutathionylation. For most proteins that undergo glutathionylation, this increases in response to elevations in reactive oxygen and nitrogen species (ROS/RNS) [36]. Thus, elevations in the aortic smooth muscle NO activate SERCA2 via Cys674 glutathionylation and increase sarcoplasmic reticulum Ca^2+^ [37]. Despite elevations in β-cell ROS and nitric oxide (NO) during glucose and lipid metabolism, the glutathionylation of the β-cell proteome is decreased in response to glucose stimulation [38]. This corresponds with an increase in the level of reduced glutathione (GSH) to oxidized glutathione (GSSG). While the mechanism for the glucose reduction in glutathiolynation remains to be determined, the low β-cell antioxidative capacity and the reduction in subcellular ROS by glucose have been proposed [38,39,40]. The impact of the glucose control of β-cell SERCA activity through reversible glutathionylation remains to be determined.

Dysfunction: Given the critical role of SERCA in Ca^2+^_ER_ storage and Ca^2+^_C_ handling, perturbations in SERCA activity and/or expression cause β-cell dysfunction (Figure 2). Islet SERCA3 expression is significantly reduced in rodent models of T2D (Goto-Kakizaki rats) [21] and T1D (non-obese diabetic (NOD)) [41]. Although SERCA3 ablation does not result in islet ER stress, how alterations in SERCA3 levels or activity impact islet function under diabetogenic stress remains to be determined. Islet SERCA2b mRNA and protein are also reduced in response to diabetogenic conditions, including (1) cytokine-treatment [11,42], (2) rodent models of T1D (NOD) or T2D (*db/db*) [19,21,41], and (3) humans with T2D [22]. This has been modeled with β-cell specific SERCA2 knockout (βS2KO) islets that show reduced Ca^2+^_ER_ uptake, depleted Ca^2+^_ER_ stores, an increased duration of the first-phase glucose-induced Ca^2+^ influx, and reduced second phase Ca^2+^ influx [43]. Furthermore, βS2KO islets display elevated ER stress and impaired proinsulin processing [43]. Under diabetic conditions resulting from exposure to a high-fat diet (HFD), diminished Ca^2+^_ER_ storage in islets with SERCA2b haploinsufficiency causes elevated ER stress, leading to impaired insulin processing and production, decreased GSIS, and increased β-cell death [10]. Importantly, the treatment of mouse models of T2D (e.g., *ob/ob* and *db/db)* with CDN1163, an SERCA2 agonist, improves glucose tolerance and reduces insulin demand [9,44]. Moreover, SERCA2 activation in β-cells (e.g., INS-1 and/or MIN6) prevents Ca^2+^_ER_ depletion and diminishes ER stress under diabetic-like conditions [10,45]. Therefore, diminished SERCA expression detrimentally affects β-cell function by disrupting Ca^2+^_ER_ handling, whereas activating SERCA holds promise for restoring Ca^2+^_ER_ stores and preserving β-cell function under diabetogenic stress.

Not only does reduced SERCA expression contribute to β-cell dysfunction, but alterations in SERCA activity also contribute to changes in β-cell Ca^2+^_ER_ handling under diabetic conditions. Elevations in O-GlcNAcylation during hyperlipidemia have been proposed to activate SERCA2 and amplify insulin secretion, which may serve as a protective measure for initially increasing Ca^2+^_ER_ stores during insulin resistance [34]. Importantly, O-GlcNAcylation-deficient islets display impaired palmitate-induced insulin secretion, which can be restored with SERCA2 activation (via CDN1163) [34]. Elevated oxidative stress during the progression of β-cell failure and diabetes onset could potentially lead to the irreversible oxidation of SERCA; this would prevent the glutathionylation-mediated activation of SERCA [46]. SERCA inhibition also occurs in response to dyslipidemia [47,48]. The exact mechanism for dyslipidemia-mediated β-cell SERCA inhibition remains to be determined; however, palmitate metabolism and changes in ER membrane lipid composition may be involved [12,49,50]. For example, palmitate-treated β-cells have increased ceramide biosynthesis, and ceramide has been shown to inhibit SERCA activity in a carcinoma cell line [49]. In addition, elevations in ER phosphatidylcholine (PC, a major component of the ER membrane) in obese mice inhibited hepatocyte SERCA activity, which can be partially restored by reducing PC [50]. Finally, insulin resistance and the associated loss of IRS-1 interaction with SERCA3 would be predicted to increase SERCA3 activation. Indeed, this is supported by IRS-1 knockout β-cells that have elevated Ca^2+^_ER_ stores and are also resistant to palmitate-induced ER stress [30]. Taken together, SERCA activity is adjusted in response to many signaling modalities that change during the pathogenesis of diabetes; the activation of SERCA supports β-cell health by increasing Ca^2+^_ER_ stores, whereas the inhibition of SERCA acutely promotes insulin secretion but chronically leads to ER stress and β-cell destruction.

### 2.2. Ryanodine Receptor (RyR)

Function: There are three RyR subunits (RyR1-3) that form functional channels in a tetrameric conformation [51]. RyRs are activated by Ca^2+^, leading to Ca^2+^_ER_ release and the activation of neighboring RyRs, which shapes the upstroke of Ca^2+^-induced Ca^2+^ release (CICR). CICR serves important roles in tuning islet hormone secretion and is therefore tightly controlled not only by the Ca^2+^-induced activation of RyRs but also by the Ca^2+^-induced closure of these receptors; Ca^2+^ binding to a high-affinity site activates RyRs, whereas Ca^2+^ binding to a low-affinity site inhibits RyRs [52]. Furthermore, the binding of certain nucleotides (e.g., cyclic AMP and ATP) allosterically increases RyRs’ open probability [53,54,55]. The tightly tuned activation and closure of RyRs play a role in orchestrating physiological β-cell Ca^2+^ signaling, which tunes insulin secretion and the β-cell ER stress response [56,57]. Although transcriptome studies show a low transcript abundance of RyRs in mouse and human β-cells, RyR1 and RyR2 expression is found in human and mouse β-cells via qRT-PCR and western blot [25,56,57,58,59,60]. Also, the activation of RyRs with ryanodine under subthreshold glucose concentrations (6 mM) resulted in Ca^2+^_C_ bursting in mouse β-cells within pancreatic slices, whereas at stimulatory glucose concentrations (8mM), the inhibition of RyRs reduced mouse β-cell Ca^2+^ oscillations, suggesting that CICR contributes to islet Ca^2+^ handling [59]. β-cell RyR2-mediated Ca^2+^_ER_ release may also amplify store-operated Ca^2+^ entry (SOCE); this was demonstrated in an RyR2-deficient β-cell line that showed reduced SOCE following SERCA inhibition. Therefore, it is important to investigate human β-cell-specific RyR functions under physiological and diabetogenic conditions, as they likely serve a role in Ca^2+^_ER_ release and potentially the ER stress response.

Dysfunction: Perturbations in RyR expression and/or function during the pathogenesis of diabetes have been shown to play a role in disrupting Ca^2+^_ER_ and cytoplasmic Ca^2+^ handling. Under conditions of protein misfolding and associated ER stress, RyR1 expression is significantly increased in INS-1, leading to Ca^2+^_ER_ release, enhanced β-cell UPR, and increased apoptosis [56,61] (Figure 2). Thus, the inhibition of RyRs in INS-1 cells and mouse islets prevents tunicamycin-induced Ca^2+^_ER_ depletion in β-cells [56]. While RyRs promote Ca^2+^_ER_ release under tunicamycin-induced ER stress, RyRs do not modulate Ca^2+^_ER_ in response to inflammatory cytokines [56]. In islets from T2D donors and *ob/ob* mice, RyR2 activity is increased by post-translational nitrosylation and oxidation [57]. Furthermore, palmitate reduces human islet sorcin expression, resulting in less of an inhibitory interaction of sorcin with RyRs [56,62]. Additionally, diabetic mouse and human islets show reduced RyR2 interaction with calstabin2, which increases the open probability by destabilizing the closed state [57]. This enhances Ca^2+^_ER_ leak and islet ER stress in *ob/ob* mice, which can be reduced with Rycal S107, a small-molecule stabilizer of calstabin2-RyR interactions [57]. Due to the longevity of β-cells, the RyR-mediated enhancement of β-cell UPR may also help protect β-cells from destruction. Future studies with human islets under diabetogenic stress are important in elucidating the contribution of elevated RyR activity to β-cell UPR and associated dysfunction and/or protection from destruction.

### 2.3. Inositol Trisphosphate Receptor (IP_3_R)

Function: The primary Ca^2+^_ER_ release channels of the β-cell are mediated by IP_3_Rs, which include three subfamily members (IP_3_R1-3; Figure 2). IP_3_R activity is modulated by several ligands, the most important of which are IP_3_ and Ca^2+^. IP_3_ increases the open probability of IP_3_Rs and modulates channel sensitivity to intracellular Ca^2+^; low Ca^2+^ levels augment IP_3_-mediated IP_3_R activation, whereas high Ca^2+^ levels promote IP_3_-mediated IP_3_R inactivation [63]. Phospholipase C (PLC) produces IP_3_ and is activated by Gq-GPCRs (e.g., primarily muscarinic M_3_ receptor and free fatty acid receptor 1 (FFAR1) in β-cells) [64,65]. Interestingly, β-cell specific muscarinic M_3_ receptor knockout mice show blunted GSIS as well as muscarinic M_3_ receptor agonist stimulated insulin secretion [66], whereas a mouse model with the β-cell specific overexpression of muscarinic M_3_ receptors shows enhanced plasma insulin without any effects on isolated islet GSIS but enhanced muscarinic M_3_ receptor agonist stimulated insulin secretion [66]. How the loss of muscarinic M_3_ receptors impacts β-cell GSIS without ligand activation remains to be determined [66]. Similarly, other Gq-GPCRs such as FFAR1 stimulate insulin secretion in an IP_3_R-dependent mechanism. The potentiation of Ca^2+^_C_ induced by FFAR1 activation was significantly reduced in MIN6 cells with the knockdown of IP_3_R [17]. Furthermore, a decrease in insulin secretion was observed in isolated mouse islets treated with both the IP_3_R inhibitor xestospongin C and the FFAR1 activator fasiglifam [17]. These findings indicate that IP_3_R activity is necessary for Gq-GPCR signaling. Furthermore, IP_3_Rs undergo post-translational modifications including phosphorylation that stimulates Ca^2+^_ER_ release. The cyclic AMP activation of protein kinase A (PKA) phosphorylates and enhances β-cell IP_3_Rs, leading to the amplification of CICR independently of RyRs [67]. IP_3_ levels are also elevated in response to the activation of specific PLCs by Ca^2+^, which may further enhance CICR. Indeed, elevated β-cell Ca^2+^ increases PLC activation, such as during 1) KCl-mediated Ca^2+^ influx [68] and 2) PLC activation occurring in phase with glucose-induced Ca^2+^ oscillations in mouse islets [69]. Taken together, the complexity and tight regulation of IP_3_Rs support its important roles in Ca^2+^_ER_ release and β-cell signaling.

Dysfunction: Changes in the expression of Gq-GPCRs and/or IP_3_Rs under diabetogenic stress significantly impact insulin secretion via changes in Ca^2+^_ER_ handling. The expression of muscarinic M_3_ receptors and, consequently, IP_3_R-mediated insulin secretion is significantly reduced in models of diabetogenic stress including *ob/ob* mice as well as mouse and human islets incubated under hyperglycemic conditions [70]. β-cell-specific muscarinic M_3_ receptor overexpressing mice are protected against HFD-induced hyperglycemia and maintain a normal glucose tolerance [66]. Furthermore, the treatment of islets from HFD-treated mice with a positive allosteric modulator (PAM) of the muscarinic M_3_ receptor significantly improved insulin secretion [71]. Interestingly, it has been reported that mouse and human islets under diabetogenic stress can induce a receptor switch from Gs to Gq-GPCR signaling pathways [72]. This implies that IP_3_R-mediated Ca^2+^_ER_ release is upregulated under diabetic conditions, despite a reduction in muscarinic M_3_ receptors, to enhance insulin secretion via Gq-GPCR signaling. However, it is important to consider the various mechanisms of IP_3_R regulation (e.g., PLC and phosphorylation) and how they are altered during diabetes pathogenesis. Enhanced IP_3_R activity and/or expression would reduce Ca^2+^_ER_ stores and may contribute to β-cell dysfunction. Indeed, ROS activates IP_3_Rs and is increased under diabetogenic conditions [73]. For example, dyslipidemia promotes oxidative stress, leading to the activation of IP_3_Rs and Ca^2+^_ER_ depletion [12,74]. IP_3_R expression has also tentatively been shown to be upregulated in T2D human islets, *db/db* mice, and HFD-fed mice [75] (Figure 2). Moreover, the treatment of INS-1 cells with cytokines and high glucose significantly reduced Ca^2+^_ER_ release stimulated by IP_3_ due to depleted Ca^2+^_ER_ stores [56]. Therefore, enhanced IP_3_R-mediated Ca^2+^_ER_ release promotes β-cell dysfunction under diabetogenic stress by diminishing Ca^2+^_ER_ stores, promoting ER stress, and blunting insulin secretion.

### 2.4. Ca^2+^_ER_ Leak Channels

Ca^2+^ homeostasis is not only modulated by ligand-gated Ca^2+^_ER_ release (e.g., RyR and IP_3_R) but also by constitutive Ca^2+^_ER_ leak. Ca^2+^_ER_ levels are set by the amount of Ca^2+^_ER_ leak and SERCA-mediated Ca^2+^_ER_ uptake; thus, Ca^2+^_ER_ storage is depleted via Ca^2+^_ER_ leak when SERCA is inhibited. Ca^2+^_ER_ leak plays an important role in tuning β-cell Ca^2+^ handling. For example, during glucose-induced Ca^2+^ oscillations that are responsible for pulsatile insulin secretion, Ca^2+^_ER_ leak occurs following plasma membrane hyperpolarization between slow waves of depolarization; this results in a reduced rate of Ca^2+^_C_ decay at the termination of each Ca^2+^ oscillation [15]. Thus, the blockade of SERCA and the depletion of Ca^2+^_ER_ stores cause a significantly accelerated return to the baseline for each glucose-induced Ca^2+^ oscillation. [15]. β-cell Ca^2+^ leak channels remain to be conclusively determined; however, two potential candidates include Sec61 of the translocon complex [76] and presenilin-1 [77,78] (Figure 2).

During the transport of proteins across the ER membrane through Sec61, Ca^2+^_ER_ leak also occurs. Specifically, Ca^2+^_ER_ leak can occur through the Sec61 channel when it is attached to a ribosome and when it is transiently open after a peptide chain and ribosome have been released [79,80]. Sec61-mediated Ca^2+^_ER_ leak is an important mechanism in regulating β-cell Ca^2+^_ER_. This is supported in human islets treated with the Sec61 agonist puromycin, which displayed significantly reduced Ca^2+^_ER_ stores [76]. Moreover, the anisomycin inhibition of Sec61 in palmitate-treated human islets enhanced Ca^2+^_ER_ storage, reduced ER stress, and improved insulin secretion [76]. Sec61-mediated Ca^2+^_ER_ leak is also modulated by the binding of Sec61 with GRP78 (also referred to as immunoglobulin heavy chain binding protein (BiP)) and calmodulin (intracellular Ca^2+^ sensing protein). In HeLa cells, the silencing of GRP78 promoted Ca^2+^_ER_ leak, and the combined silencing of GRP78 and Sec61 abolished this effect [81]. Therefore, GRP78 limits Ca^2+^_ER_ leak and promotes Ca^2+^_ER_ storage in part due to its interaction with Sec61. This is further supported by a mutation in the Sec61α subunit (Y344H) that prevents GRP78 binding and increases Ca^2+^_ER_ leak through Sec61 [81,82]. Interestingly, when this mutation is introduced in mice, diabetes develops as a result of exacerbated ER stress and β-cell apoptosis. The expression of wild-type Sec61α in the mutant mice prevented HFD-induced hyperglycemia and reduced ER stress [83]. Thus, it is likely that a disruption of Sec61 interaction with GRP78 contributes to Ca^2+^_ER_ store depletion, ER stress, apoptosis, and diabetes. Sec61 also contains an IQ calmodulin binding motif in its N-terminus. Calmodulin inhibited Sec61 currents in a Ca^2+^-dependent manner in rough microsome vesicles [84]. When calmodulin was inhibited in HeLa cells, an increase in Ca^2+^_ER_ leak was observed, and this was prevented by the knockdown of Sec61α [84]. Taken together, these studies support Sec61 as a Ca^2+^_ER_ leak channel that is regulated in a Ca^2+^-dependent manner. The reduced Ca^2+^-dependent inhibition of Sec61 by GRP78 and/or calmodulin under conditions of ER stress enhances Ca^2+^ leak and promotes β-cell failure.

Presenilin-1 and presenilin-2 have also been shown to form Ca^2+^_ER_ leak channels. For example, mouse embryonic fibroblasts (MEFs) with the ablation of presenilin-1 and presenilin-2 displayed significantly reduced Ca^2+^_ER_ leak (~80%), which was restored with the expression of either presenilin-1 or presenilin-2 [85]. However, this is controversial, and another group has also shown that presenilin-1 and presenilin-2 ablated MEFs display no changes in Ca^2+^_ER_ [86]. In β-cells, presenilin-1 has been identified as a component in Ca^2+^_ER_ handling and hypoxia-induced cell death. Presenilin-1’s Ca^2+^_ER_ leak function is activated by glycogen synthase kinase 3 β (GSK3β) phosphorylation. The knockdown of presenilin-1 or the inhibition of GSK3β significantly reduced glucose-induced Ca^2+^ oscillations, first-phase insulin secretion, mitochondrial ATP production, and respiration in β-cells [77,78]. Although these findings suggest that presenilin-1-mediated Ca^2+^_ER_ leak plays a role in the concerted regulation of β-cell function, future studies with ER-localized Ca^2+^ indicators are required to confirm if and how presenilins are involved in Ca^2+^_ER_ leak and storage [86,87].

### 2.5. Two-Pore Domain K^+^ Channel (K2P)

ER Function: A subset of K2P channels are localized and functional on the ER membrane, where they provide a K^+^ countercurrent during Ca^2+^_ER_ release [88,89] (Figure 2). ER K2P currents prevent negative charge buildup on the intraluminal membrane following Ca^2+^_ER_ release. β-cells express two ER-localized K2P channels including TALK-1 (*KCNK16*) and TASK-1 (*KCNK3*). *KCNK16* is the most highly expressed K^+^ channel in human β-cells. The knockout of mouse β-cell *kcnk16* or the dominant negative inhibition of human β-cell TALK-1 channels result in increased Ca^2+^_ER_ storage due to reduced Ca^2+^_ER_ release [88]. ER TALK-1 channel K^+^ flux helps to dissipate ER membrane potential from being generated during Ca^2+^_ER_ release. If the ER membrane potential were to become hyperpolarized, it would move towards the equilibrium potential of Ca^2+^ and thus reduce the electrical driving force for Ca^2+^ across the ER membrane. The TALK-1 augmentation of Ca^2+^_ER_ release also activates Ca^2+^-activated K^+^ currents (termed K_slow_) that hyperpolarize the membrane potential between Ca^2+^ oscillations [88]. Thus, the TALK-1 enhancement of Ca^2+^_ER_ release slows glucose-induced Ca^2+^ oscillation frequency. Altogether, β-cell K2P channels play a role in tuning Ca^2+^_ER_ handling, Ca^2+^_C_, and insulin secretion. Importantly, a single-nucleotide polymorphism (SNP) that results in a gain-of-function (GOF) in TALK-1 is associated with an increased risk for developing T2D and may impair β-cell Ca^2+^_ER_ handling during the pathogenesis of diabetes [90].

Dysfunction: The β-cell ablation of TALK-1 not only reduces the ER stress response under diabetogenic conditions but also limits glucose intolerance in response to a diabetogenic HFD [88,90]. This predicts that overactive TALK-1 channels may exacerbate ER stress and contribute to glucose intolerance during the pathogenesis of diabetes. To date, there are two coding sequence (CDS) changes in TALK-1 that causes GOF; these include the T2D-associated nonsynonymous SNP (rs1535500) in *KCNK16* as well as a monogenic mutation in TALK-1 (L114P) that causes maturity onset diabetes of the young (MODY) [91,92]. GOF TALK-1 channels localized to the ER would be predicted to increase Ca^2+^_ER_ release and diminish Ca^2+^_ER_ storage. Indeed, there is significant reduction in Ca^2+^_ER_ storage following the expression of TALK-1 L114P and increased IP_3_-induced Ca^2+^_ER_ release in β-cells from a mouse model containing the TALK-1 L114P mutation [92]. How these alterations of Ca^2+^_ER_ handling caused by TALK-1 L114P contribute to β-cell dysfunction remains to be determined. However, TALK-1 L114P also localizes to the plasma membrane, where it prevents glucose-induced Ca^2+^ influx by hyperpolarizing the membrane potential; this is primarily responsible for blunted GSIS and severe glucose intolerance in TALK-1 L114P mice [92]. The expression of the other TALK-1 GOF channel in β-cells (resulting from rs1535500, TALK-1 A277E) caused a reduction in Ca^2+^_ER_ stores and elevated the ER stress response to tunicamycin [88]. Not only does rs1535500 lead to a GOF polymorphism in TALK-1, but it is also in strong linkage disequilibrium with SNPs in the promoter and coding regions of the downstream gene *KCNK17* (TALK-2), resulting in increased expression and GOF [93]. In addition to high extracellular pH, TALK-1 and, to a greater extent, TALK-2 are activated by long-chain CoAs, ROS, and nitric oxide in heterologous systems [94,95], all of which are elevated in diabetes. Interestingly, TALK-1 and TALK-2 form functional heterodimers [96]. Moreover, TALK-1/TALK-2 heterodimers have been detected in the human β-cell line Endo-Cβh5 [96]. The modulation of β-cell Ca^2+^_ER_ handling by TALK-2 and/or TALK-1/TALK-2 heterodimers remains to be determined. Taken together, the increased activity of TALK channels may contribute to β-cell dysfunction under diabetogenic stress by enhancing Ca^2+^_ER_ depletion and the associated ER stress response.

### 2.6. Store-Operated Ca^2+^ Entry (SOCE)

Function: When Ca^2+^_ER_ levels decrease, a mechanism is activated to help replenish Ca^2+^_ER_ stores, which is termed store-operated Ca^2+^ entry (SOCE). SOCE primarily involves the Ca^2+^ sensing stromal interaction molecule 1 (STIM1), Orai1 Ca^2+^ channels, and transient receptor potential canonical channel 1 (TRPC1) [97,98]. When Ca^2+^_ER_ levels are at homeostasis under basal conditions, STIM1 EF hands bind to Ca^2+^, leading to dimerization and diffuse distribution throughout the ER membrane. A reduction in Ca^2+^_ER_ leads to the oligomerization of STIM1 dimers at ER/plasmalemmal junctions, where STIM1 interacts with and activates Orai1 and TRPC1 channels, stimulating extracellular Ca^2+^ influx, which is then pumped into the ER via SERCA [98,99]. SOCE was significantly impaired in INS-1 cells that expressed dominant negative mutants of either Orai1 or TRPC1 and the knockout of STIM1; SOCE was measured following Ca^2+^_ER_ depletion by SERCA inhibition (with thapsigargin) or muscarinic M_3_ receptor-mediated IP_3_R activation [100,101]. GSIS was reduced in mouse and rat islets treated with pharmacological SOCE inhibitors, which reduce glucose-induced Ca^2+^ influx [100,101]. Long-chain free fatty acids also induce β-cell SOCE following FFAR1-mediated IP_3_R activation [17]. MIN6 cells treated with the FFAR1 agonist fasiglifam displayed enhanced intracellular Ca^2+^ and subsequent insulin secretion that was blunted when STIM1 or Orai1 was silenced or when IP_3_R was inhibited [17]. In mice with conditional β-cell specific STIM1 knockdown, SOCE induced by SERCA inhibition or FFAR1 activation was significantly impaired [17]. Furthermore, in vivo studies show that insulin secretion was not altered without STIM1 during FFAR1 activation, whereas an increase in insulin secretion was observed in the controls. This supports that STIM1, and thus SOCE, is necessary for the FFAR1 potentiation of insulin secretion [17]. Altogether, β-cell SOCE contributes to glucose and free fatty acid stimulated insulin secretion; thus, dysfunction in any of the SOCE components would be expected to impact β-cell Ca^2+^ handling and ER stress.

Dysfunction: SOCE is impaired in diabetic settings and promotes β-cell dysfunction. STIM1 mRNA and protein levels are significantly reduced in human islets from T2D donors, islets from hyperglycemic mice (low-dose streptozotocin-treated), and cytokine and palmitate-treated INS-1 cells [101]. This suggests the reduced STIM1-mediated activation of SOCE and Ca^2+^ depletion under diabetogenic stress. Indeed, chronic hyperglycemic (72 h) conditions reduced SOCE in INS-1 cells [100]. However, tunicamycin-induced ER stress resulted in mouse islet Ca^2+^ oscillations at subthreshold glucose concentrations, which was abolished using SOCE inhibitors [102]. T2D donor islets with STIM1 overexpression show enhanced GSIS, likely due to enhanced SOCE [101]. Because most conditions of β-cell stress reduce Ca^2+^_ER_ stores, the resulting SOCE may play an important role in the initial β-cell response to stress and eventual dysfunction.

## 3. Mutations That Affect Ca^2+^_ER_ Function and Ca^2+^ Handling

### 3.1. RyR2 Mutations

GOF mutations in RyR2 cause β-cell dysfunction and glucose intolerance. For example, RyR2 mutations (e.g., R2474S or N2386I) that disrupt the binding of calstabin2 result in leaky RyR2 channels [103] (Figure 3). These RyR2 mutations result in catecholaminergic polymorphic ventricular tachycardia (CPVT), glucose intolerance, and impaired insulin secretion [57]. Furthermore, mouse models with leaky RyR2 channels (with R2474S, N2386I, or S2814D mutations) displayed increased β-cell Ca^2+^_ER_ release with a concomitant elevation in basal Ca^2+^_C_ [57,104]. Leaky RyRs reduce Ca^2+^_ER_ storage, causing ER stress and reduced GSIS. Ca^2+^_ER_ release is also coupled with mitochondrial Ca^2+^ uptake, which is essential for several Ca^2+^-dependent energy processes such as ATP production (discussed in the next section). Thus, β-cells from mice expressing RyR R2474S or N2386I show defects in the mitochondrial structure and upregulation in markers of mitochondrial dysfunction, including uncoupling protein 2 and peroxisome proliferator-activator receptor γ coactivator 1a [57]. Importantly, reducing leaky RyR2s (R2474S or N2386I) with Rycal S107 restored Ca^2+^_ER_ levels, enhanced insulin secretion, and improved mitochondria abnormalities [57]. Altogether, GOF mutations in RyR2 disrupt β-cell function by reducing Ca^2+^_ER_ stores, which causes ER and mitochondrial stress, resulting in impaired insulin secretion and glucose intolerance.

### 3.2. Wolfram Syndrome 1 (WFS1) Mutations

Wolfram syndrome 1 (WFS1) is a rare autosomal recessive genetic disorder characterized by diabetes insipidus, diabetes mellites, optic atrophy, and deafness [105,106,107]. The *WFS1* gene encodes the protein wolframin, an ER localized glycoprotein that has been linked to β-cell ER stress, Ca^2+^_ER_ handling, and proteostasis. To date, there are over 200 mutations in *WSF1* that result in a loss of function (LOF) and cause WFS1 [105,108]. Patients with two LOF *WSF1* mutations show diabetic phenotypes, and carriers with one *WSF1* LOF allele show an increased risk for developing T2D [109]. Within the islet, *WFS1* is abundant in β-cells and has little to no expression in α, δ, or pancreatic polypeptide cells [110]. To mimic WFS1 LOF, *Wfs1* knockout mouse models were utilized. *Wfs1*-deficient mice displayed reduced glucose-induced Ca^2+^ influx and impaired insulin secretion; GSIS was restored with the overexpression of *Wfs1* [110]. The knockdown of *Wfs1* in INS-1 cells caused Ca^2+^_ER_ depletion [111]. Additionally, *Wfs1*-deficient β-cells promote cytokine-induced ER stress by upregulating proinflammatory cytokines and chemokines (Figure 3). Hyperglycemia-induced ER stress in *Wfs1* knockout INS-1 cells involves PERK upregulation [112]. This suggests that *Wfs1* is critical for limiting ER stress in β-cells under inflammatory or hyperglycemic conditions. Defects in Ca^2+^_ER_ handling can be restored in β-cells with WFS1 deficiency following calpain inhibition (calpain initiates apoptosis) [113]. Taken together, WFS1 plays a crucial role in maintaining β-cell Ca^2+^_ER_ and reducing ER stress; thus, diabetes observed in patients with WFS1 LOF mutations likely involves β-cell dysfunction due to reduced Ca^2+^_ER_ and ER stress.

### 3.3. Insulin Mutations

Insulin biosynthesis accounts for up to 50% of glucose-stimulated β-cell protein production. Conditions of increased insulin demand (e.g., insulin resistance) and elevated insulin production result in ER stress, protein misfolding, and activation of the UPR [14]. The UPR is an important mechanism that protects β-cells from ER stress by reducing protein production, removing misfolded proteins, and increasing chaperone abundance. However, several insulin mutations lead to unresolvable ER stress; this occurs due to the buildup of insulin molecules in the ER by impairing insulin processing or via elevated insulin production due to reduced insulin receptor activation (Figure 3). To date, over 70 *INS* gene mutations have been identified; these mutations result in various diabetic phenotypes such as neonatal diabetes and mutant insulin-gene-induced diabetes of the youth (MIDY) [114,115,116,117]. Many *INS* mutations expressed in MIN6 cells show ER retention and cause ER stress, monitored by *Chop* upregulation [118]. *INS* mutations have been shown to disrupt Ca^2+^ handling. For example, glucose-induced Ca^2+^ influx was nearly abolished in islets from a mouse model of the C96Y insulin mutation (Akita mouse model); this mutation disrupts the disulfide bond formation between the A and B chains of insulin and proinsulin folding and causes insulin retention in the ER [56,119]. Interestingly, the ryanodine inhibition of RyR partially restored glucose-induced Ca^2+^ oscillations in islets from the Akita mouse, which suggests that C96Y leads to overactive RyRs and Ca^2+^_ER_ depletion [56]. Another ER-retained *INS* mutation A24D also reduced Ca^2+^_ER_ in a dose-dependent manner, and treatment with either translation inhibitors or the chemical chaperone TUDCA reduced Ca^2+^_ER_ depletion [111,118,120]. Therefore, pathogenic *INS* mutations elevate ER stress and impair Ca^2+^_ER_ handling, which causes β-cell dysfunction and, ultimately, failure.

## 4. Ca^2+^_ER_ in the Context of Cellular Function

### 4.1. Chaperone-Mediated Ca^2+^ Binding

The majority of Ca^2+^_ER_ is bound to chaperone proteins including calnexin, calreticulin, protein disulfide isomerase (PDI), glucose-regulated protein 78 (GRP78, also known as BiP), and glucose-regulated protein 94 (GRP94). Most of these chaperones have a high capacity for Ca^2+^ due to multiple low-affinity Ca^2+^ binding sites and thus tune Ca^2+^_ER_ homeostasis [121]. As part of the ER stress response, these chaperones are upregulated to improve protein folding and maintain Ca^2+^_ER_ levels. The roles of β-cell chaperone function have begun to emerge and have been well established in other cells. The C-terminal domain of calreticulin is responsible for buffering up to 50% of Ca^2+^_ER_, and thus, the overexpression of calreticulin increases Ca^2+^_ER_ stores [121,122,123]. SOCE is also reduced in calreticulin overexpressing mouse L-fibroblasts due to increased Ca^2+^_ER_ storage [124]. β-cell Ca^2+^_ER_ depletion during NO-mediated apoptosis was reduced through the overexpression of calreticulin [125]. Therefore, the calreticulin upregulation observed in β-cells under diet-induced stress may serve to enhance Ca^2+^_ER_ storage, reduce ER stress, and delay β-cell failure [126]. The glucose-regulated proteins were first characterized by their upregulation in cells following glucose starvation [127]. Two of the most abundant GRPs in β-cells are GRP78 and GRP94. GRP78 has been shown to sequester up to 25% of Ca^2+^_ER_ [121,128]. Owing in part to its Ca^2+^_ER_ buffering capacity, β-cell GRP78 overexpression protected mice from HFD-induced insulin resistance and ER stress [129]. Increasing levels of Ca^2+^_ER_ alter the conformation of GRP94 to increase peptide binding [130]. GRP94-deficient cells show greater ER stress following Ca^2+^_ER_ depletion, but the loss of GRP94 itself did not induce ER stress or disrupt Ca^2+^_ER_ homeostasis [131]. The Ca^2+^_ER_ buffering capabilities of ER chaperones help maintain Ca^2+^_ER_ homeostasis and prevent Ca^2+^_ER_ depletion under stress.

### 4.2. Protein Folding

Ca^2+^ is also necessary for chaperone-mediated protein folding [132]. Thus, a reduction in Ca^2+^_ER_ impairs the protein folding capabilities of GRP78, GRP94, and PDIs; this leads to protein misfolding and the initiation of the UPR [133]. The knockdown of β-cell GRP78 or GRP94 increased UPR, reduced insulin content, and limited GSIS [134,135]. GRP78 and GRP94 are upregulated during β-cell ER stress to promote protein folding and Ca^2+^_ER_ storage [135,136]. Thus, increasing the INS-1 levels of GRP78 reduced hyperglycemia-mediated UPR activation [134]. GRP94 interacts with proinsulin, promoting its processing, which increases the granule content of insulin and amplifies GSIS [135]. As mature insulin includes three disulfide bonds, the formation and cleavage of these disulfide bonds by PDIs also play an important role in insulin processing. A mouse model harboring the biallelic LOF mutation in *Pdia6* encoding PDIA6 showed decreased β-cell insulin content, resulting in hyperglycemia [137], whereas the overexpression of *P4HB* encoding PDI promoted ER stress and impaired GSIS, which is likely due to the accumulation of proinsulin in the ER resulting from the PDI cleavage of proinsulin disulfide bonds [134]. Indeed, the knockdown of PDI in INS-1 cells elevated insulin disulfide bond formation and exit from the ER [138]. The activity of certain PDIs can be modulated by interactions with other chaperones. For example, the localization of PDI1A is dependent on Ca^2+^_ER_; the depletion of Ca^2+^_ER_ promotes the interaction of PDI1A with calreticulin and the inhibition of its chaperone function [139]. Taken together, many chaperones efficiently orchestrate the multifaceted aspects of insulin processing. These chaperones also help amplify insulin production in response to increased insulin demand and reduce ER stress by limiting insulin misfolding and/or increasing Ca^2+^_ER_ storage.

### 4.3. Mitochondrial Function

The ER and mitochondria interact at mitochondrial-associated membranes (MAMs), which enables the transfer of lipids and Ca^2+^ between these organelles. The MAM proteins involved in Ca^2+^ flux include GRP75, IP_3_R, and voltage-dependent anion channel (VDAC). GRP75 connects IP_3_Rs on the ER membrane with VDAC1 on the outer mitochondrial membrane [140]. Acute glucose stimulation increases IP_3_R2/VDAC1 interactions in INS-1 cells, elevating both mitochondrial Ca^2+^ content and the Gq-GPCR-mediated uptake of mitochondrial Ca^2+^ during IP_3_R2 Ca^2+^_ER_ release [141] (Figure 4). The knockdown of Grp75 reduced IP_3_R2/VDAC1 interactions, limited mitochondrial Ca^2+^ uptake during IP_3_R-mediated Ca^2+^_ER_ release, and blunted GSIS [141]. In T2D patients, β-cell IP_3_R2/VDAC1 interactions are reduced, which likely limits mitochondrial Ca^2+^ influx and contributes to β-cell dysfunction [142].

Both ER-stress-mediated Ca^2+^_ER_ depletion and increased Ca^2+^_ER_ storage can increase mitochondrial Ca^2+^. During chronic hyperglycemia, Ca^2+^_ER_ depletion leads to increased basal mitochondrial Ca^2+^ uptake, which disrupts human islet glucose metabolism and GSIS [141]. Thus, ER stress initiates a greater transfer of Ca^2+^_ER_ to the mitochondria; this leads to mitochondrial dysfunction, due in part to decreased ATP synthase activity, elevated ROS generation, and increased fission [141] (Figure 4). Alternatively, increasing Ca^2+^_ER_ stores via SERCA activation increased mitochondrial Ca^2+^ uptake and ATP synthesis as well as prevented mitochondrial dysfunction in palmitate-treated β-cells [45]. The β-cell specific knockdown of TALK-1 channels also enhanced Ca^2+^_ER_ storage, which led to increased mitochondrial Ca^2+^, ATP synthesis, and insulin secretion [143]. Therefore, alterations in Ca^2+^_ER_ directly influence mitochondrial Ca^2+^, which either impairs β-cell function via mitochondrial Ca^2+^ overload or improves β-cell function via enhanced ATP production and subsequent insulin secretion.

## 5. Conclusions

Ca^2+^_ER_ plays an important role in modulating β-cell function and the response to stress. During insulin resistance and β-cell depletion preceding type 2 and type 1 diabetes, respectively, increased insulin production leads to Ca^2+^_ER_ depletion. Reduced Ca^2+^_ER_ enhances β-cell ER stress and promotes the UPR. As β-cells are long-lived, the Ca^2+^_ER_ regulation of the stress response also serves to modulate the workload and prevent or delay β-cell failure [144]. An integrated signaling network precisely orchestrates Ca^2+^_ER_ handling. The altered expression and/or activity of proteins involved in Ca^2+^_ER_ handling modifies insulin production and secretion, in part through changes in the Ca^2+^_ER_ release into cytosol and/or mitochondria. Moreover, changes in Ca^2+^_ER_ are sensed by many proteins (e.g., STIMs and chaperones) that modulate many aspects of β-cell function, including Ca^2+^_ER_ refilling, transcription, translation, and protein processing. Therefore, mutations that perturb Ca^2+^_ER_ result in β-cell dysfunction and monogenic diabetes. Due to the critical role that Ca^2+^_ER_ depletion plays in β-cell dysfunction, increasing Ca^2+^_ER_ could be used to restore β-cell function and health in diabetes. However, Ca^2+^_ER_ homeostasis is important to most cells, and many of the proteins that control Ca^2+^_ER_ handling are ubiquitously expressed. Thus, future studies are required to identify the β-cell-specific aspects of Ca^2+^_ER_ handling that can be targeted without deleterious effects on other tissues.

## Figures and Tables

**Figure 1 biology-13-00379-f001:**
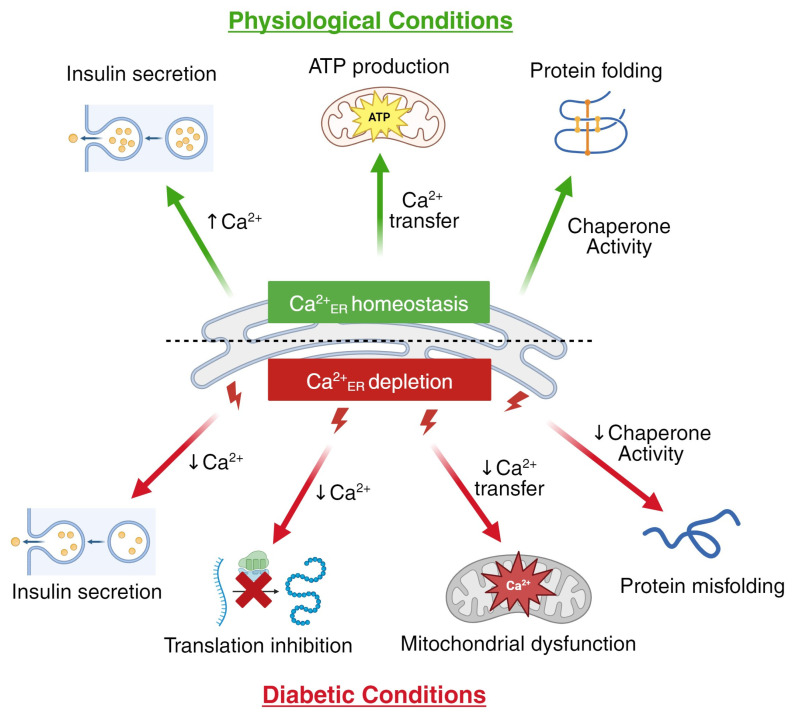
Ca^2+^_ER_ is essential for β-cell function and becomes disrupted under diabetic conditions. The precise level and release of Ca^2+^ from the ER modulates chaperone function, promotes protein folding, augments insulin secretion, and contributes to mitochondrial ATP generation via Ca^2+^ transfer (upper panels, green). The depletion of Ca^2+^_ER_ stores observed in type 1 and type 2 diabetes is detrimental to β-cells, initiating blunted insulin secretion, reduced translation, protein misfolding, and mitochondrial Ca^2+^ oversaturation, resulting in dysfunction (lower panels, red). Arrows denote the direction of change with physiological or diabetic conditions. Created with BioRender.com (accessed on 28 March 2024).

**Figure 2 biology-13-00379-f002:**
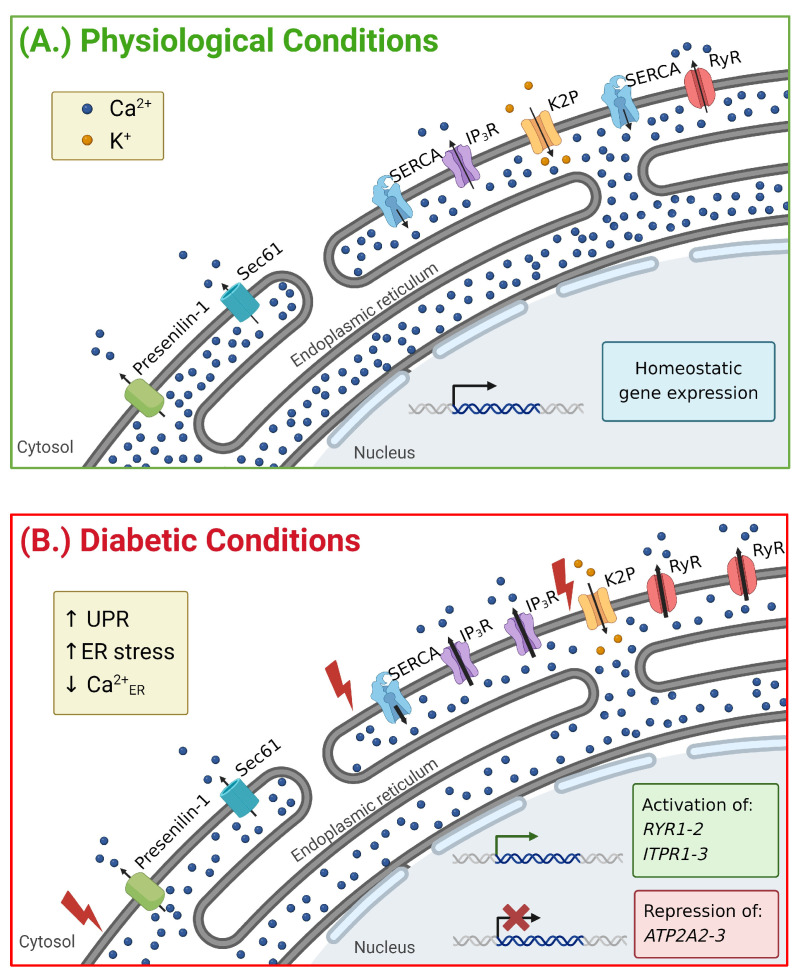
β-cell Ca^2+^_ER_ handling proteins under physiological and diabetogenic stress. (**A**) Ca^2+^_ER_ handling proteins maintain the high concentration of Ca^2+^_ER_ to support ER function. SERCA pumps Ca^2+^ against its concentration gradient to fill Ca^2+^_ER_ stores. IP_3_R and RyR release Ca^2+^_ER_ in response to various stimuli and contribute to increasing Ca^2+^_C_ and insulin secretion. β-cell Ca^2+^_ER_ is also modulated by Ca^2+^ leak channels that remain to be determined but may include Sec61 and/or presenilin-1. (**B**) Under diabetogenic stress, the activity and expression of Ca^2+^_ER_ handling proteins are altered. The increased expression of IP_3_R and RyR promotes Ca^2+^_ER_ depletion and ER stress. Additionally, SERCA pumps are downregulated, which limits Ca^2+^_ER_ uptake. This depletion in Ca^2+^_ER_ stores is observed in diabetes and contributes to β-cell dysfunction. The activity of RyR and IP_3_R is increased under diabetogenic conditions and contributes to Ca^2+^_ER_ depletion. Although SERCA expression is reduced with diabetogenic conditions, activity is likely increased to preserve Ca^2+^_ER_ stores. Arrows denote movement of Ca^2+^ (blue) or K^+^ (orange) through indicated channels. Thicker arrows show increased channel activity. Created with BioRender.com (accessed on 28 March 2024).

**Figure 3 biology-13-00379-f003:**
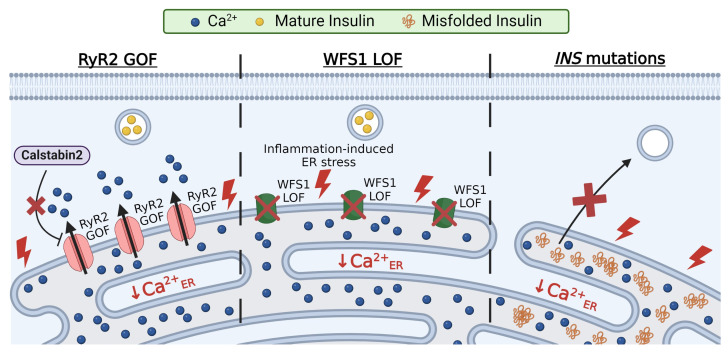
Monogenic mutations that impair β-cell Ca^2+^_ER_. RyR2 mutations that disrupt the binding of its endogenous inhibitor, calstabin2, result in RyR2 GOF; this enhances Ca^2+^_ER_ release and reduces GSIS (movement of Ca^2+^ through RyR2 denoted by arrows; **left panel**). WFS1 is critical for limiting β-cell ER stress under proinflammatory conditions; thus, the loss of WFS1 function results in inflammation-induced ER stress, diminished Ca^2+^_ER_ stores, and blunted insulin secretion (**middle panel**). Mutations in the *INS* gene disrupt insulin trafficking and lead to insulin aggregation and accumulation in the ER; this results in β-cell dysfunction, in part due to chronic ER stress, UPR, and depleted Ca^2+^_ER_ stores (**right panel**). Created with BioRender.com (accessed on 28 March 2024).

**Figure 4 biology-13-00379-f004:**
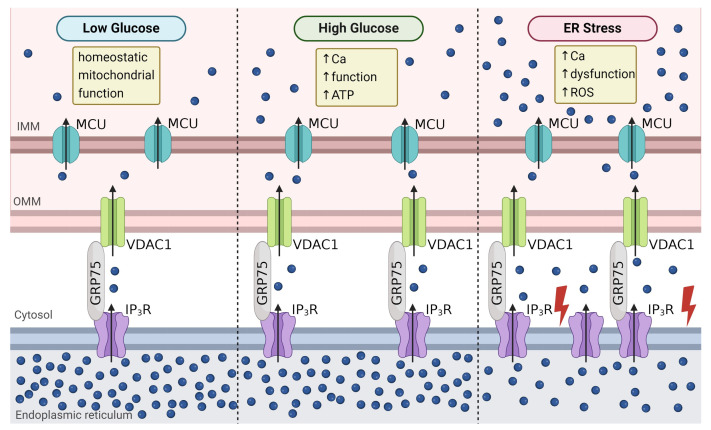
β-cell mitochondrial-associated membranes. Under low glucose conditions, VDAC1/IP_3_R/GRP75 interactions help maintain mitochondrial Ca^2+^ (**left panel**). Upon glucose stimulation, an increase in VDAC1/IP_3_R/GRP75 interactions raises mitochondrial Ca^2+^ levels. This transfer of Ca^2+^ into the mitochondria stimulates ATP synthesis and insulin secretion (**middle panel**). IP_3_R-mediated Ca^2+^_ER_ depletion under ER stress increases the uptake of mitochondrial Ca^2+^. This Ca^2+^ overload in the mitochondrial enhances ROS generation and impairs function (**right panel**). Arrows indicate the movement of Ca^2+^ through channels. Abbreviations: IMM—inner mitochondrial membrane; OMM—outer mitochondrial membrane; MCU—mitochondria Ca^2+^ uniporter (teal); VDAC1- voltage-dependent anion channel (green); IP_3_R- inositol trisphosphate receptor (purple); ROS—reactive oxygen species. Created with BioRender.com (accessed on 28 March 2024).

## Data Availability

Not applicable.

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
