# Peer review of "Disrupted Endoplasmic Reticulum Ca2+ Handling: A Harβinger of β-Cell Failure"

_biology, 2024, doi:10.3390/biology13060379_

Round 1

Reviewer 1 Report

Comments and Suggestions for Authors

The manuscript "Disrupted endoplasmic reticulum Ca2+ handling: a harbinger of b-cell failure" focuses on the mechanisms that regulate ER Ca2+ homeostasis and their role in the impairment of insulin secretion. 

The review includes enough information to address the aim of the review clearly and deeply. 

Minor observations: 

Title: substitute the greek "beta" letter by "b" in "harbinger"

In line 129, the authors refer to mouse models and mention Goto-Kakizaki rats. It is suggested to change "mouse models" to "rodent models"

Line 137: it is suggested to include "increased duration of the first phase glucose-induced Ca2+ influx" to clarify. Additionally, eliminate reference 38 in the same line.

Line 144: The authors mention that "As changes in b-cell secretion under diabetogenic stress promote insulin resistance, SERCA2 activation may improve insulin sensitivity in part by restoring the kinetics of b-cell insulin secretion". However, it is not clear how does the insulin secretion process could be linked to insulin sensitivity in other tissues or how does decreased insulin secretion may promote insulin resistance.

Authors may consider to eliminate the sentence.

Line 306: The figure legend stands that "the activity of RyR and IP3R are increased under diabetogenic conditionas and contribute to Ca2+ER depletion" and "SERCA expression is reduced with diabetogenic conditions, activity is likely increased to preserve Ca2+ER stores" are not shown. The authors could represent the activity of the "calcium transporters" by adjusting the size of the arrows. 

Line 347: authors could add "coding sequence (CDS) changes"

Line 355: Substitute "where is prevents" for "where it prevents"

Line 523: change "and inhibition its chaperone function" to "and inhibition of its chaperone function"

Suggestions: 

ER calcium handling consists on a critical process involved not only in insulin secretion but also in beta-cell survival. It could be useful that the authors include some paragraphs in the introduction section in order to detail the physiologic role of ER Ca+ in these processes during normal conditions.

Author Response

Response to Reviewer Comments:

Reviewer 1

The manuscript "Disrupted endoplasmic reticulum Ca2+ handling: a harbinger of b-cell failure" focuses on the mechanisms that regulate ER Ca2+ homeostasis and their role in the impairment of insulin secretion.

The review includes enough information to address the aim of the review clearly and deeply.

We would like to thank the Reviewer for their time in reviewing our manuscript. We appreciate the helpful feedback and assistance in ensuring the highest quality publication possible.

Minor observations:

In line 129, the authors refer to mouse models and mention Goto-Kakizaki rats. It is suggested to change "mouse models" to "rodent models"

We thank the reviewer for this suggestion and have revised the manuscript accordingly.

Line 137: it is suggested to include "increased duration of the first phase glucose-induced Ca2+ influx" to clarify. Additionally, eliminate reference 38 in the same line.

We thank the reviewer for this suggestion and have revised the manuscript accordingly.

Line 144: The authors mention that "As changes in b-cell secretion under diabetogenic stress promote insulin resistance, SERCA2 activation may improve insulin sensitivity in part by restoring the kinetics of b-cell insulin secretion". However, it is not clear how does the insulin secretion process could be linked to insulin sensitivity in other tissues or how does decreased insulin secretion may promote insulin resistance.

Authors may consider to eliminate the sentence.

We thank the reviewer for this suggestion and have revised the manuscript accordingly.

Line 306: The figure legend stands that "the activity of RyR and IP3R are increased under diabetogenic conditionas and contribute to Ca2+ER depletion" and "SERCA expression is reduced with diabetogenic conditions, activity is likely increased to preserve Ca2+ER stores" are not shown. The authors could represent the activity of the "calcium transporters" by adjusting the size of the arrows.

We thank the reviewer for this suggestion and have revised the arrows accordingly.

Line 347: authors could add "coding sequence (CDS) changes"

We thank the reviewer for this suggestion and have revised the manuscript accordingly.

Line 355: Substitute "where is prevents" for "where it prevents"

We thank the reviewer for pointing out this typo and have revised the manuscript accordingly.

Line 523: change "and inhibition its chaperone function" to "and inhibition of its chaperone function"

We thank the reviewer for pointing out this typo and have revised the manuscript accordingly.

Title: substitute the greek "beta" letter by "b" in "harbinger"

The reviewer notes a witty substitution of β for b in harβinger. This substitution was intentional to make readers understand that this review foreshadows β-cells as the subject. This was done in part to draw attention to this review and hope that this substitution will help draw more interest. We also believe that readers of this title will understand the intention of harbinger.

Suggestions:

ER calcium handling consists on a critical process involved not only in insulin secretion but also in beta-cell survival. It could be useful that the authors include some paragraphs in the introduction section in order to detail the physiologic role of ER Ca+ in these processes during normal conditions.

We thank the reviewer for these suggestions and have added two paragraphs to the introduction detailing these important roles for beta-cell ER calcium (see 1.2 and 1.3).

Reviewer 2 Report

Comments and Suggestions for Authors

The review by Dobson and Jacobson was a pleasure to read and fits the journals scope. The manuscript summarized the current state of calcium in the ER of pancreatic beta cells appropriatly and comprehensively. The authors focus not just on health and disease implications of calcium in the ER but extend their summary to show current trends and future directions of interest in the field. The review will be greatly appreciated by peers as a valuable ressource summarizing the state of the art and pinpointing novel directions.

Minor comments:

Diabetes is commonly defined as a glucocentric problem although decades of research clearly display the importance of dyslipidemia in addition to hyperglycemia. While this review appropriatly distinguishes different models/patholgoies driving calcium dysfunction in the ER of beta cells, dyslipdemia seems to be underrepresented. Extending or specfically mentioning which phenotypes dyslipdemia causes in pancreatic beta cell calcium homeostais could improve the quality of the work presented.

No further comments.

Author Response

Reviewer 2     

The review by Dobson and Jacobson was a pleasure to read and fits the journals scope. The manuscript summarized the current state of calcium in the ER of pancreatic beta cells appropriatly and comprehensively. The authors focus not just on health and disease implications of calcium in the ER but extend their summary to show current trends and future directions of interest in the field. The review will be greatly appreciated by peers as a valuable ressource summarizing the state of the art and pinpointing novel directions.

We would like to thank the Reviewer for their time in reviewing our manuscript. We appreciate the helpful feedback and assistance in ensuring the highest quality publication possible. We have addressed all the reviewer’s comments and revised the manuscript accordingly. These changes have strengthened the manuscript and are summarized below.

Minor comments:

Diabetes is commonly defined as a glucocentric problem although decades of research clearly display the importance of dyslipidemia in addition to hyperglycemia. While this review appropriatly distinguishes different models/patholgoies driving calcium dysfunction in the ER of beta cells, dyslipdemia seems to be underrepresented. Extending or specfically mentioning which phenotypes dyslipdemia causes in pancreatic beta cell calcium homeostais could improve the quality of the work presented.

We thank the reviewer for this suggestion and have now added details of dyslipidemia in the context of ER calcium throughout the manuscript (these changes can be found highlighted and by searching the document for dyslipidemia).